Anatomical and biomechanical traits of broiler chickens across ontogeny. Part I. Anatomy of the musculoskeletal respiratory apparatus and changes in organ size

Tickle Peter G. 1
Paxton Heather 2
Rankin Jeffery W. 2
Hutchinson John R. 2
Codd Jonathan R. 1 jonathan.codd@manchester.ac.uk
1 Faculty of Life Sciences, University of Manchester , Manchester , UK
2 Structure & Motion Laboratory, Department of Comparative Biomedical Sciences, The Royal Veterinary College, University of London , Hatfield, Hertfordshire , UK
Yang Xiang-Jiao
Electronic publication date: 2014 Jul 3
Publication date: 2014
Volume: 2
Electronic Location ID: e432
Received 2014 Apr 1; Accepted 2014 May 28
Copyright: © 2014 Tickle et al.
Copyright year: 2014
Copyright holder: Tickle et al.
License: This is an open access article distributed under the terms of the Creative Commons Attribution License, which permits unrestricted use, distribution, reproduction and adaptation in any medium and for any purpose provided that it is properly attributed. For attribution, the original author(s), title, publication source (PeerJ) and either DOI or URL of the article must be cited.
License URL: https://creativecommons.org/licenses/by/4.0/

Keywords: Broiler, Development, Breathing, Organ, Pathology, Scaling

Funding: BBSRC grant BB/I021116/1 This work was supported by BBSRC grant BB/I021116/1 to JRC and JRH. The funders had no role in study design, data collection and analysis, decision to publish, or preparation of the manuscript.

==============================
Genetic selection for improved meat yields, digestive efficiency and growth rates have transformed the biology of broiler chickens. Modern birds undergo a 50-fold multiplication in body mass in just six weeks, from hatching to slaughter weight. However, this selection for rapid growth and improvements in broiler productivity is also widely thought to be associated with increased welfare problems as many birds suffer from leg, circulatory and respiratory diseases. To understand growth-related changes in musculoskeletal and organ morphology and respiratory skeletal development over the standard six-week rearing period, we present data from post-hatch cadaveric commercial broiler chickens aged 0, 2, 4 and 6 weeks. The heart, lungs and intestines decreased in size for hatch to slaughter weight when considered as a proportion of body mass. Proportional liver size increased in the two weeks after hatch but decreased between 2 and 6 weeks. Breast muscle mass on the other hand displayed strong positive allometry, increasing in mass faster than the increase in body mass. Contrastingly, less rapid isometric growth was found in the external oblique muscle, a major respiratory muscle that moves the sternum dorsally during expiration. Considered together with the relatively slow ossification of elements of the respiratory skeleton, it seems that rapid growth of the breast muscles might compromise the efficacy of the respiratory apparatus. Furthermore, the relative reduction in size of the major organs indicates that selective breeding in meat-producing birds has unintended consequences that may bias these birds toward compromised welfare and could limit further improvements in meat-production and feed efficiency.

Introduction

Genetic selection in domesticated broiler chickens has brought about significant improvements in the form of increasing meat yields and growth performance. Growth rates in intensively reared industrial broiler chickens have consistently accelerated such that a 300% increase has been engineered in the past 60 years, from 25 g per day in the 1950s to 100 g per day in the modern bird (Knowles et al., 2008). Consequently the optimal slaughter mass of approximately 3 kg is reached in six rather than 16 weeks (Griffin & Goddard, 1994; Havenstein et al., 1994a; Govaerts et al., 2000). Maximising pectoral (breast) muscle mass is a primary target for selection. Compared to ancestral varieties, pectoral hypertrophy in the broiler chicken has resulted in an approximate doubling in muscle size, making up ∼20% of total body mass in the modern bird (Havenstein, Ferket & Qureshi, 2003b; Schmidt et al., 2009). Mounting evidence suggests that selection for such economically desirable traits in the modern broiler has been accompanied by reduced welfare (Julian, 1998; Knowles et al., 2008) and increased mortality (Havenstein et al., 1994a; Havenstein, Ferket & Qureshi, 2003a). Considerable research is being directed toward understanding welfare problems such as the multitude of leg pathologies that may affect locomotion in broiler chickens (Kestin et al., 1992; Bradshaw, Kirkden & Broom, 2002; Corr et al., 2003a; Corr et al., 2003b; Knowles et al., 2008; Paxton et al., 2010), cardiac (Wilson, Julian & Barker, 1988) and pectoral (Randall, 1982) myopathies, pulmonary hypertension (Wideman, 2001) and ascites (Wilson, Julian & Barker, 1988; Julian, 1993). The prevalence of these conditions indicates that further improvements in industry-efficiencies and meat production may be constrained by the physiological capabilities of broilers because skeletal, cardiac, respiratory and digestive systems appear to be close to their functional limit.

Characterising the relationships between body mass and organ, skeleton and muscle size are crucial to our understanding of animal physiology. Physical scaling rules determine the structural and functional consequences of changes in size and therefore exert a profound effect on organismal form (Schmidt-Nielsen, 1984). Understanding the relative growth and size of organs and muscles is important in broilers as it can help to better understand the diseases that they suffer from. For example, the broiler heart and brain become progressively smaller as a proportion of body mass over development, unlike in ancestral breeds (Jackson & Diamond, 1996; Schmidt et al., 2009). In contrast, selection for faster growth and muscle mass is reflected in the proportionally greater intestine mass and accelerated pectoral growth in broiler chickens (Jackson & Diamond, 1996; Konarzewski et al., 2000; Schmidt et al., 2009). These relationships demonstrate how artificial selection in the broiler has resulted in developmental trade-offs; reallocation of resources to maximize nutrient absorption and pectoral muscle mass has coincided with a relative decrease in the size of other organs (Havenstein et al., 1994b; Jackson & Diamond, 1996; Havenstein, Ferket & Qureshi, 2003b; Schmidt et al., 2009). To better understand the effects of intensive selection on broilers, more information is required on how increasing body mass and growth rate has shaped their anatomy and physiology across ontogeny. Therefore, in this paper we present data on how organ and muscle growth varies with increasing body mass in a commercial broiler strain.

Broiler chickens suffer from respiratory problems that may be related to their rapid musculoskeletal development (Julian, 1998) and their potential to outgrow pulmonary capacity (Wideman, 2001). The avian respiratory system can be considered as a two-part mechanism comprising a pump (musculoskeletal) and gas exchanger (lung). The primary ventilatory mechanism in birds consists of dorsal and ventral movements of the ribs and sternum (Zimmer, 1935; Claessens, 2009) that affect air sac volume, thereby facilitating a unidirectional flow of air through the lung (Bretz & Schmidt-Nielsen, 1971; Scheid & Piiper, 1971). Uncinate processes, which are small bones that extend caudodorsally from the vertebral ribs, have been shown to function as levers that assist rib and sternal movements during breathing (Tickle et al., 2007). Respiratory movements of the skeleton generate pressure changes within the thorax that are necessary to drive inhalation and exhalation, both of which are active processes, driven by respiratory muscle activity (Codd et al., 2005). Respiratory muscle activity requires metabolic energy consumption. However, studies in guinea fowl (Numida meleagris) (Markley & Carrier, 2010) have demonstrated that breathing constitutes only 2% of whole-organism metabolism (Markley & Carrier, 2010). In contrast, research on load carrying (Tickle, Richardson & Codd, 2010) and behavioural energetics (Tickle, Nudds & Codd, 2012) indicate a higher cost of breathing in barnacle geese. In particular, carrying extra weight on the sternum, analogous to increased pectoral muscle mass seen in broiler chickens, is energetically expensive compared to an equivalent weight carried on the back (Tickle, Richardson & Codd, 2010). Furthermore, barnacle geese have been shown to achieve metabolic savings by changing posture; when compared to sitting, standing is associated with a 25% higher resting metabolic rate (Tickle, Nudds & Codd, 2012). The higher cost of standing has been in part attributed to the energetic cost of moving the heavy weight of the sternum with each breath, which does not occur while sitting (Tickle, Nudds & Codd, 2012). Understanding movements of the sternum are important when considering how selection has shaped morphology in domestic birds due to the selection for enhanced pectoral growth in meat-producing domestic ducks (Gille & Salomon, 1998; Maruyama, Akbar & Turk, 1999), turkeys (Swatland, 1979) and broiler chickens (Havenstein et al., 1994b; Govaerts et al., 2000; Havenstein, Ferket & Qureshi, 2003b; Schmidt et al., 2009).

Only limited information is available on the developmental biology of the avian respiratory skeleton. In the domestic turkey (Meleagris gallopavo) a trend for delayed ossification, especially in the uncinate processes, was identified that potentially constrained respiratory performance due to the decreased rigidity of cartilaginous compared to fully ossified bones; cartilaginous uncinate processes will yield under muscle strain before comparable ossified bone would, making them less effective levers (Tickle & Codd, 2009). Furthermore, the growth trajectory of respiratory muscles in the broiler is unknown. Scaling of respiratory muscle growth in proportion to overall body size and, perhaps most importantly when we consider the dorso-ventral breathing movements of the sternum, the pectoral muscles is likely a key factor in determining how effectively and efficiently breathing functions in broiler chickens. To better understand the respiratory problems apparent in broiler chickens, an analysis of how the skeleton develops is necessary. Here we present a description of developmental changes in the musculoskeletal elements of the broiler chicken respiratory system together with an evaluation of how organ size scales with increasing body mass.

Materials and Methods

Specimens

Broiler chickens from a popular commercial strain were obtained from a commercial supplier. Birds were sampled in a post-hatch growth series between days 1 and 42 (i.e., weeks 0–6), corresponding to a 50 × range in total body mass (Mb) (Table 1).

Table 1 Internal organ growth over development.

Anatomical and biomechanical traits of broiler chickens across ontogeny. Internal organ mass as a proportion of total body mass (Mb) over development. Data are mean ± standard deviation.

Age (days)	n	Mb (kg)	Heart (% Mb)	Lung (% Mb)	Liver (% Mb)	Intestine (% Mb)	
1	10	0.04 ± 0.003*	0.74 ± 0.05	1.04 ± 0.20*	2.99 ± 0.34*	17.1 ± 2.71*	
14	10	0.59 ± 0.06*	0.71 ± 0.10	0.75 ± 0.16*	3.63 ± 0.30	9.89 ± 1.05*	
28	9	1.40 ± 0.09*	0.66 ± 0.06	0.72 ± 0.09*	3.10 ± 0.32	8.27 ± 0.90*	
42	13	3.27 ± 0.21*	0.53 ± 0.07*	0.55 ± 0.12*	2.48 ± 0.22	5.34 ± 0.51*	
Notes.

* Significant differences at the 0.05 level.

Musculoskeletal growth

Mb, pectoralis major (pectoralis), pectoralis minor (supracoracoideus) and external oblique muscle mass (Mm), wing and ribcage mass, fibre length (Lf) and pennation angle (θ) were measured using an electronic balance (±0.001 g), ruler (±1 mm) and protractor (± 1°). To account for variation within muscle architecture, Lf and θ were calculated as the mean of five measurements in each muscle. Physiological cross-sectional area (PCSA) was calculated for each muscle (Eq. (1); Sacks & Roy, 1982): (1) PCSA=Mm∗cosθ/ρ∗Lf.

Density of muscle tissue (ρ) was assumed to be 1.06 g cm−3 (Mendez & Keys, 1960; Paxton et al., 2010). Linear measurements of the sternum and uncinate processes were recorded using a ruler (±1 mm). Average length of the uncinate processes was calculated from those occurring on ribs 2–5 since these processes were found in all specimens. Girth was measured around the circumference of the thorax, tucked under the wings. One-way ANOVA was used to test for differences in morphology, using mean values of specimens from each developmental stage. Justification for using parametric analysis was based upon the results of a Shapiro–Wilk test and the normal quantile–quantile plot that assessed assumptions of data normality (results displayed in Appendix).

The scaling relationships between musculoskeletal characters and body mass were determined using reduced major axis (RMA) linear regression, a method that is suitable since it takes into account variation in both x and y axes (Rayner, 1985; Sokal & Rohlf, 1995). All regression analyses were performed on log10-transformed data to establish allometric equations in the form: (2) logy=loga+b logx

where a represents the intercept and the exponent b represents the slope of the line equation. Upper and lower 95% confidence intervals (CI) and the R2 value were calculated for each regression line slope. RMA analyses were performed in the PAST statistical program (Hammer, Harper & Ryan, 2001). Assuming geometric similarity (i.e., isometry) over ontogeny, all dimensions are expected to scale in proportion to each other meaning that lengths should scale to Mb0.33, areas to Mb0.67 and masses to Mb1.00. Isometric scaling was assumed where the regression slope ±95% CI overlapped the expected value.

Histological staining

The ossification pattern of the thoracic skeleton over ontogeny was examined using the histochemical staining protocol of Tickle & Codd (2009). All muscle tissue was removed using dissection, the preparation cleaned by immersion in a 1% potassium hydroxide (KOH) solution and then the skeleton was treated with solutions of alcian blue (uptake corresponds to cartilage) and alizarin red (uptake corresponds to bone). Photographs of stained specimens were taken using a light microscope (Leica MZ9s; Leica Microsystems, Milton Keynes, UK) and subsequently analysed in Leica image analysis software. For comparison of structural properties, relative area of bone and cartilage was calculated for the uncinate process that projects from the fourth vertebral rib in all specimens (Tickle & Codd, 2009).

Results

Organ development

Heart and lung mass follow a negative allometric growth pattern, decreasing relative to body mass over the six-week growth period (Tables 1 and 2; Fig. 1A). Heart mass decreases from 0.74% to 0.53% of body mass over the growth period, while proportional lung mass reduces by almost half, decreasing from 1.04% to 0.55% (Tables 1 and 2; Fig. 1B). Proportional liver mass significantly increased between 0 and 2 weeks, reached a peak of 3.63% on day 14, then decreased between 2 and 6 weeks when it accounted for 2.48% of Mb (Table 1; Fig. 1C). Taking all data into account indicated that overall liver mass followed an isometric growth pattern; i.e., in direct proportion to increasing Mb (Table 2). Repeating the scaling analysis for birds only aged between 0 and 2 weeks found positive allometric growth, Mb1.10 while birds aged between 2 and 6 weeks had negative allometric growth, Mb0.76 (Table 2; Fig. 1C). Total intestine mass was found to strongly decline as a proportion of Mb over growth with a negative allometric regression slope of Mb0.75 (Tables 1 and 2; Fig. 1D).

Table 2 Scaling relationship between organ and body mass.

The regression slope indicates the proportional change of organ mass with increasing body mass, and 95% confidence intervals are shown (95% CL). Isometric (=) and negative allometric (−) growth are indicated by symbols.

	N	Slope	Lower
95% CL	Upper
95% CL	R 2	
Heart	69	0.91 (−)	0.89	0.94	0.98	
Lungs	62	0.86 (−)	0.83	0.90	0.97	
Liver (wks. 0–6)	69	0.95 (=)	0.92	1.00	0.98	
Liver (wks. 0–2)	30	1.10 (+)	1.06	1.15	0.99	
Liver (wks. 2–6)	59	0.76 (−)	0.72	0.80	0.95	
Intestine	69	0.75 (−)	0.72	0.78	0.98	

Figure 1 Breast muscle growth during development.

Scatterplots showing log transformed pectoralis major and minor masses against log body mass over development from 0 to 2-weeks (solid line) and 2–6-weeks old (dashed line). The equations describing lines of best fit were: (A) pectoralis major 0–2 weeks: Mb1.83–3.28 (n = 30, R2 = 0.99; p < 0.001); 2–6 weeks: Mb1.29–1.83 (n = 59; R2 = 0.98; p < 0.001). (B) pectorals minor 0–2 weeks: Mb1.87–4.06 (n = 30, R2 = 0.98; p < 0.001); 2–6 weeks: Mb1.28–2.48 (n = 58; R2 = 0.96; p < 0.001).

Carcass parts

No significant difference was detected between proportional wing masses with increasing age, indicating a directly proportional relationship with Mb. After accounting for variation due to body mass, girth significantly increased during growth (Table 3), reflecting a relative lateral expansion of the thorax. In contrast, as a proportion of Mb, ribcage mass was significantly lower at day 28 than at days 14 or 42 (Table 3) whereas normalise d keel length displayed a trend for increased length, being highest in 42-day birds (Table 3).

Table 3 Morphological examination of the external body.

Data represented are mean ± standard deviation. Following the principles of geometric similarity (Alexander et al., 1981), girth and keel length are normalised by body mass1/3 to negate the effect of body size on our data.

Age (days)	n	Wings (% Mb)	Girth	Keel length	Rib cage (% Mb)	
14	10	7.5 ± 0.6	0.21 ± 0.01*	0.08 ± 0.004*	10.0 ± 1.1	
28	9	7.8 ± 0.5	0.23 ± 0.01*	0.07 ± 0.004*	7.49 ± 0.9*	
42	12	5.8 ± 0.4	0.25 ± 0.01*	0.10 ± 0.005 *	8.9 ± 0.7	
Notes.

* Significant differences at the 0.05 level.

Thoracic anatomy

Pectoralis major and minor (i.e., M. pectoralis and M. supracoracoideus) Mm increased as a proportion of Mb over development, showing strong positively allometric growth (Tables 4 and 5). The growth of these muscles was defined by two phases: an initial rapid increase in Mm between weeks 0 and 2 was followed by a relatively slower increase between weeks 2 and 6 (Table 5; Fig. 2). Sternal keel length and depth developed with positive allometry, increasing above the expected geometric scaling exponent (Mb0.33) (Table 6), while mean uncinate process length scaled to Mb0.30, indicating reduced length with increasing body mass. Growth of external oblique muscle was found to increase in direct proportion to increasing body mass (Table 6).

Figure 2 Ossification of the thoracic skeleton.

Representative stained skeletons showing the progression of ossification in the vertebral ribs, uncinate processes and sternum. Blue areas correspond to cartilage and red areas to bone. Ossification of ribs and uncinate processes are shown for the hatchling (A and B), 2-week old (C and D) and 6-week old (E and F) birds panels. Ossification of the uncinate processes and sternum remain incomplete at slaughter age. Scale bars represent 10 mm.

Table 4 Breast muscle growth over development.

Anatomical and biomechanical traits of broiler chickens across ontogeny. Breast muscle mass (% Mb) over development. Data are mean ± standard deviation.

Age (days)	Pectoralis major (% Mb)	Pectoralis minor (% Mb)	
1	0.58 ± 0.05*	0.12 ± 0.02*	
14	8.65 ± 0.98*	1.88 ± 0.27*	
28	12.12 ± 1.12*	2.51 ± 0.31*	
42	14.50 ± 1.71*	3.10 ± 0.37*	
Notes.

* Significant differences at the 0.05 level.

Table 5 Growth of breast Mm.

Positive allometry is seen throughout development.

	Age
(weeks)	N	Slope	Lower
95% CL	Upper
95% CL	R 2	
Pectoralis major
(M. pectoralis)	0–6	69	1.60	1.56	1.66	0.98	
0–2	30	1.83	1.77	1.88	0.99	
2–6	59	1.29	1.24	1.34	0.98	
Pectoralis minor
(M. supracoracoideus)	0–6	68	1.62	1.57	1.70	0.98	
0–2	30	1.87	1.79	1.95	0.98	
2–6	58	1.29	1.22	1.36	0.96	

Table 6 Scaling relationships of thoracic musculoskeletal parameters and body mass.

The regression slope indicates proportional change with increasing body mass. Isometric (=), positive (+) and negative (−) allometric growth are indicated by symbols.

		n	Slope	Lower 95% CL	Upper 95% CL	R 2	
Pectoralis major	M m	69	1.60 (+)	1.56	1.66	0.98	
	L f	37	0.46 (+)	0.42	0.50	0.95	
	PCSA	37	1.23 (+)	1.19	1.27	0.99	
Pectoralis minor	M m	68	1.62 (+)	1.57	1.70	0.98	
	L f	34	0.55 (+)	0.50	0.62	0.89	
	PCSA	29	1.17 (+)	1.09	1.25	0.97	
External oblique	M m	25	0.97 (=)	0.84	1.09	0.89	
	L f	15	0.31 (=)	0.12	0.48	0.32	
	PCSA	15	0.90 (=)	0.72	1.09	0.86	
Sternal keel	Length	34	0.48 (+)	0.44	0.51	0.97	
	Depth	34	0.55 (+)	0.51	0.59	0.95	
Uncinate process	Length	31	0.30 (−)	0.28	0.32	0.94	

Skeletal development

Ossification of the uncinate processes commences at around the time of hatch; 15-day embryos have entirely chondrified processes whereas one-day old chicks show 40% ossification (Table 7; Fig. 3). Uncinate process bone synthesis proceeds from the midpoint in proximal and distal directions and overall bone area plateaus at ∼77% of total uncinate process area at 40 days old. The remaining 23% remains cartilaginous, shared between tip and base (Table 7). At hatch the sternum exhibits ossification in the most proximal portion in addition to centres of ossification in the caudolateral processes. While bone growth extends distally along the sternum, at slaughter age ossification of the sternal keel remains incomplete (Fig. 3).

Figure 3 Organ growth during development.

Scatterplots showing log transformed organ masses against log body mass over development from hatch to 6-weeks old. The equations describing lines of best fit were: (A) heart: Mb0.91–1.96; (n = 69; R2 = 0.98; p < 0.001); (B) lung: Mb0.86–1.76 (n = 62; R2 = 0.97; p < 0.001); (C) liver: 0–2 weeks old (solid line): Mb1.10–1.70 (n = 30; R2 = 0.99; p < 0.001); 2–6 weeks old (dashed line): Mb0.76–0.77 (n = 59; R2 = 0.95; p < 0.001); (D) intestine: Mb0.75–0.34 (n = 69; R2 = 0.98; p < 0.001).

Table 7 Ossification of the uncinate processes.

Anatomical and biomechanical traits of broiler chickens across ontogeny. Structural changes in uncinate process 4 over development from embryo (6 days before hatch) to slaughter age; presence of cartilage and bone as derived from stained specimens. Data presented as mean ± standard deviation (SD).

	Bone	Cartilage at base	Cartilage at tip	
Age (days)	% total area	SD	% total area	SD	% total area	SD	
−6	0	0	100	0	100	0	
1	39.5	16.8	35.0	9.4	25.5	9.6	
13	72.6	8.4	23.5	8.1	3.9	0.2	
29	72.6	7.4	13.4	7.7	14.0	0.3	
40	76.8	8.6	9.1	5.1	14.2	2.5	

Discussion

Organ development

Our observations confirm a decrease in heart and lung mass relative to body mass over development (Tables 2 and 3). These findings mirror the reduction in relative heart (Havenstein et al., 1994b; Govaerts et al., 2000; Thaxton, 2002; Havenstein, Ferket & Qureshi, 2003b; Schmidt et al., 2009) and lung mass (Havenstein et al., 1994b; Govaerts et al., 2000; Havenstein, Ferket & Qureshi, 2003b) seen over ontogeny in Ross-type broiler breeds. While absolute mass of the heart is higher than found in unselected lines, proportional mass is lower at hatching and the difference progressively increases over development (Schmidt et al., 2009). Similarly, lung mass progressively declines as a proportion of body mass over development in our broiler strain, mirroring the negative allometric lung growth in Ross broilers (Mb0.84 compared to Mb0.86 in this study) (Govaerts et al., 2000). Considering our results together with previous reports indicates that reduced circulatory and respiratory capacity is an unintended consequence of genetic selection for rapid growth and high Mb. The reduction in broiler heart and lung mass compared to slower-growing, lighter breeds can be considered to be a contributing factor in the increased mortality and disease in modern broilers (Wideman, 2001; Havenstein, Ferket & Qureshi, 2003b). Further developments in genetic selection for increasing growth and improving breast yields may be constrained by limited respiratory and circulatory functional capacity. The increased incidence of physiological problems and mortality (Havenstein et al., 1994a; Havenstein, Ferket & Qureshi, 2003a) in modern broilers indicates that these systems are already working close to maximal levels to satisfy the physiological demands of growth.

A complex pattern of liver growth was revealed, with organ mass growing proportionally faster than overall body mass growth in the two weeks after hatch, followed by a proportional decrease in mass between two weeks and six weeks post-hatch. Liver development in Ross-type broilers follows a similar pattern of proportionally decreasing after a period of rapid growth soon after hatch (Govaerts et al., 2000; Schmidt et al., 2009). Furthermore, Ross broiler peak liver mass as a proportion of Mb was reached on day 7 (3.80%) (Schmidt et al., 2009) and day 8 (3.38%) (Govaerts et al., 2000), although the findings presented here do not include birds sampled at these stages. Nevertheless, proportional liver mass at day 14 is similar between our study’s and Ross birds (3.15% (Govaerts et al., 2000); or approximately 3.5% (Schmidt et al., 2009) of Mb). The relatively fast pace of growth in the first two weeks, followed by proportional decline, indicates that the liver matures rapidly, ready for the transition from fat-rich yolk stores to a predominantly carbohydrate diet (Schmidt et al., 2009). Genetic selection for digestive efficiency in broilers may directly influence the developmental profile of the liver due to the important function it plays in carbohydrate and fat metabolism.

Total intestine mass declined relative to Mb during growth, which is similar to previous reports of proportional decline in intestine mass over development in Ross-type broilers (Iji, Saki & Tivey, 2001; Schmidt et al., 2009). Absolute and proportional values of the intestine are higher in this study, potentially in part due to a larger intestine in this commercial broiler strain and also in part due to methodological differences; here we report intestine mass and contents rather than the empty intestine mass (Iji, Saki & Tivey, 2001). Exhaustion of the yolk sac contents presumably accounts for the large (6.9%) decline in proportional intestine mass during the first 14 days of post hatch development (Viera & Moran, 1999). After this period, relative intestine mass decreases by approximately 2–2.5% every 14 days (Table 1), which is similar to previous reports (calculations using the data provided in Table 2 of Iji, Saki & Tivey (2001) and Fig. 5 of Schmidt et al. (2009) indicate an approximate decrease of 2.4%). Schmidt et al. (2009) reported complicated post-hatch intestinal growth, with positive allometric growth in the first seven days followed by a sustained period of negative allometric growth, possibly reflecting an early maturation of the digestive system (Govaerts et al., 2000). To balance this relative mass loss and maintain high feed efficiency, morphometric changes occur during this period to increase surface area for absorption of nutrients from ingested food (Schmidt et al., 2009).

Carcass parts

While proportional wing mass did not significantly change over growth the overall average yield of 7.0% is below that reported in Ross broilers (8.6%: Havenstein et al., 1994b; 7.9%: Havenstein, Ferket & Qureshi, 2003b). The absence of significant change in wing mass is consistent with the absence of flight as a locomotor mode and the reliance on leg muscles for terrestrial locomotion, and occurs despite steep increases in pectoral muscle mass. Girth in 42-day-old broilers was very similar to a recent report for pureline and broiler birds (Paxton et al., 2013). Presumably the rapid increase in pectoral muscle mass is a factor in the reduced proportional ribcage mass over development. Normalised sternal keel length was found to increase over growth, indicating that the area available for pectoral muscle attachment is greater as the birds reach slaughter weight, consistent with the rapid proportional increase in pectoral mass.

Thoracic anatomy

The relationship between increasing pectoral mass and body mass was positively allometric, indicating that the breast muscles became proportionally much larger over the six-week growth period. Allometric relationships were stronger than those reported for Ross broilers (Govaerts et al., 2000; Schmidt et al., 2009), indicating that the birds in this study were able to lay down breast muscle more rapidly during development. Overall breast muscle size as a proportion of Mb was similar to the values reported for Ross (Govaerts et al., 2000; Havenstein, Ferket & Qureshi, 2003b; Schmidt et al., 2009) and similar commercial strain (Paxton et al., 2013) broilers.

Allometric scaling was found for sternal keel dimensions; this proportional increase in the size of the sternum over development presumably enables fast growth of the breast muscles. In contrast, average uncinate process length was found to decrease over growth, scaling with a slight negative allometry. Uncinate processes act as levers for movement of the ribs and sternum during breathing, and overall length is an important factor that determines the magnitude of this leverage (Tickle et al., 2007). Therefore, the relative shortening of uncinate processes might correspond to a decrease in ventilatory performance during growth.

Considering that sternal mass (i.e. breast muscle, (Table 6)) increases with positive allometry during growth it was expected that the external oblique muscle would undergo similar growth to maintain functional efficacy. This muscle attaches to the base of uncinate processes and the sternal margin and is active during the exhalation phase of breathing, when it elevates the sternum dorsally (Codd et al., 2005). Our data indicate however that the external oblique muscle becomes progressively smaller in comparison to the breast muscles as the birds age, and may be less able to maintain adequate expiration. Further evidence is provided by the isometric growth of Lf and PCSA, since maximum force-generating capacity increases in direct proportion to muscle area (Lieber & Friden, 2000). To our knowledge, no comparable data on respiratory muscle development exist, so it is difficult to establish whether this pattern of growth is unique to the broiler or more widespread among ancestral fowl breeds and other species. Coupled with the decreased proportional heart and lung masses, however, selection for meat-producing traits appears to have the undesirable effect of compromising the engine of aerobic metabolism; i.e., the respiratory system. Clearly more research into the respiratory anatomy in birds is required to understand the potential trade-offs between breast growth and breathing performance.

Skeletal development

The broiler skeleton ossifies following a pattern similar to other domestic poultry (Hogg, 1980; Maxwell, 2008; Tickle & Codd, 2009). Ossification of skeletal characters, however, is seen to develop at different time points. For example, the long bones of the leg and wing are almost entirely ossified at hatch, while the sternum, tip of the scapula and proximal and distal portions of the ribs ossify after hatch (Fig. 3). Ossification of uncinate processes begins around the time of hatch, in agreement with an earlier report in the chicken (Hogg, 1980), when bone replacement of cartilage becomes apparent in the midpoint of the shaft (Table 7; Fig. 3). Rapid synthesis of bone tissue must occur in the days immediately prior to hatch, considering that the processes are entirely chondrified six days before hatch (Table 7). Onset of ossification therefore coincides with the initial in ovo respiratory skeleton movements and transition from chorioallontoic gas exchange to pulmonary ventilation (Menna & Mortola, 2002); perhaps this mechanical stimulus prompts remodeling of uncinate process structure toward ossified tissue. Uncinate process growth in the broiler is similar to development in the domestic turkey (Tickle & Codd, 2009) in that ossification remains incomplete, with a plateau in bone growth at around 80% of total process area. An interesting difference between domestic turkey and chicken is the lack of a bony symphysis between the base of the process and adjoining rib in the broiler (Tickle & Codd, 2009). A cartilaginous base may have significant implications for the functional efficacy of the uncinate process because they act as force-amplyfying levers for muscle contractions during rib movement (Tickle et al., 2007). The cartilaginous portion of uncinate processes has a lower elastic modulus, or is less stiff, than the ossified area which might compromise the lever function of the uncinate processes (Tickle & Codd, 2009). It is possible that an ossified connection between uncinate process and rib would form later in development, as is seen in the domestic turkey between days 64 and 94 post-hatch (Tickle & Codd, 2009), suggesting that immaturity of the skeleton even at slaughter weight may have a functional impact on breathing in the broiler.

Ossification of the sternum is incomplete over the six-week period, with specifically the distal portion of the keel remaining chondrified. The cartilaginous distal portion of the sternum is common to meat-producing and laying strains (Breugelmans et al., 2007) and presumably reflects the reduced or negligible importance of flight as a means of locomotion in domestic fowl; without need for a sturdy anchor to accommodate and dissipate the strong contractions of pectoral muscles during flapping flight, it seems the energetic resources are directed to growing muscle tissue rather than synthesis of bone for the sternum.

The rapid development of body mass in broilers is not mirrored by accelerated ossification of the respiratory skeleton. Instead the skeleton appears to develop in a pattern similar to other poultry, indicating that this process is tightly conserved within galliform birds. Given the potential compromise in respiratory performance due to cartilaginous skeletal elements coupled with a large sternal weight, selection for faster and increased ossification may provide a benefit to broiler health.

Here we have considered the how the growth trajectories of major organs and respiratory elements are affected by increasing body mass in broilers. In contrast to the strong pectoral growth, major organs become proportionally smaller with age and development of the respiratory pump is delayed. Our findings indicate that selection for rapid growth and large breast muscles could have the unintended consequence of negatively affecting broiler physiology by compromising the function capacity of cardiovascular and respiratory systems.

Supplemental Information

Appendix Statistical test for data normality

Results of the Shapiro-Wilk test to assess the normality of morphological data.

Click here for additional data file.

Supplemental Information 2 Raw data on anatomical properties of developing broiler chickens

Tables 1 and 2 contain the data generated by dissection of an ontogenetic sequence of broilers.

Click here for additional data file.

We are thankful to Cobb-Vantress Inc. for providing broiler chicken specimens.

Additional Information and Declarations

Competing Interests

Author Contributions

John Hutchinson is an Academic Editor for PeerJ. The authors declare there are no competing interests.

Peter G. Tickle conceived and designed the experiments, performed the experiments, analyzed the data, wrote the paper, prepared figures and/or tables, reviewed drafts of the paper.

Heather Paxton conceived and designed the experiments, performed the experiments, analyzed the data, prepared figures and/or tables, reviewed drafts of the paper.

Jeffery W. Rankin conceived and designed the experiments, performed the experiments, reviewed drafts of the paper.

John R. Hutchinson and Jonathan R. Codd conceived and designed the experiments, contributed reagents/materials/analysis tools, reviewed drafts of the paper.

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
