# Peer review of "Anatomical and biomechanical traits of broiler chickens across ontogeny. Part I. Anatomy of the musculoskeletal respiratory apparatus and changes in organ size"

_PeerJ, doi:10.7717/peerj.432_

## Round 0.1 · original submission · Minor Revisions

· Academic Editor

Minor Revisions

I am pleased to report that both reviewers have recommended acceptance of your paper. Just for best accuracy and readability, please go through the manuscript once more and revise it accordingly where it is necessary.

Reviewer 1 ·

Basic reporting

This manuscript appears technically sound. It is logically organised, and generally reads very well.

Experimental design

Overall, this manuscript presents good quality work. The investigative approach is logical and well executed.

Validity of the findings

This work appears to be based on a solid knowledge of the subject.

Comments for the author

Overall very good work. However, as a potential reader of this publication my first impression would be that this manuscript is excessively lengthy and somewhat overloaded with data sets. I would suggest to shorten the manuscript, and present the data sets, in more concise manner. For instance, the information presented in Tables 1 &2 can be fitted in a single table, as well the data presented in Tables 4, 5, and 6 can be presented in a single Table.

You have analysed (discussed) your findings in the context of the data of others in text, at length, but as a reader of your publication I would like to see some summary figure or table presenting your current numbers, and comparing the studied variables on historical time scale from the perspective of natural history of changes of these variables at different stages of genetic broiler improvement over the last 50 years or so. Undoubtedly, all readers of your work would appreciate this.

Reviewer 2 ·

Basic reporting

no comments

Experimental design

the experimental design is sound

Validity of the findings

the findings are valid

Comments for the author

This is a nicely prepared and fully referenced manuscript. A pleasure to read. I believe the DISCUSSION provides a fair overview and summary of the data.

---

## Round 0.2 · accepted · Accept

· Academic Editor

Accept

Thanks for considering and making the necessary revision.